# Adefovir Dipivoxil as a Therapeutic Candidate for Medullary Thyroid Carcinoma: Targeting RET and STAT3 Proto-Oncogenes

**DOI:** 10.3390/cancers15072163

**Published:** 2023-04-05

**Authors:** Tariq Alqahtani, Vishnu Kumarasamy, Sahar Saleh Alghamdi, Rasha Saad Suliman, Khalid Bin Saleh, Mohammed A. Alrashed, Mohammed Aldhaeefi, Daekyu Sun

**Affiliations:** 1Department of Pharmaceutical Sciences, College of Pharmacy, Ministry of National Guard Health Affairs, King Saud bin Abdulaziz University for Health Sciences, Riyadh 11481, Saudi Arabia; 2Medical Research Core Facility and Platforms, King Abdullah International Medical Research Center (KAIMRC), Ministry of National Guard Health Affairs, Riyadh 11426, Saudi Arabia; 3Department of Pharmacology and Toxicology, College of Pharmacy, University of Arizona, Tucson, AZ 85721, USA; 4Department of Molecular and Cellular Biology, Roswell Park Cancer Center, Buffalo, NY 14203, USA; 5Department of Cancer Genetics and Genomics, Roswell Park Cancer Center, Buffalo, NY 14203, USA; 6Pharmacy Department, Fatima College of Health Sciences, Almafrag, Abu Dhabi P.O. Box 3798, United Arab Emirates; 7Department of pharmacy practice, College of Pharmacy, King Saud bin Abdulaziz University for Health Sciences, Riyadh 11481, Saudi Arabia; 8Department of Clinical and Administrative Pharmacy Sciences, College of Pharmacy, Howard University, Washington, DC 20059, USA; 9The BIO5 Institute, University of Arizona, Tucson, AZ 85721, USA; 10Department of Cancer Biology, University of Arizona, Tucson, AZ 85724, USA

**Keywords:** medullary thyroid carcinoma, adefovir dipivoxil, RET, STAT3, drug design, Egr-1, cancer, EMT

## Abstract

**Simple Summary:**

In this study, we employed a cell-based assay with a luciferase gene controlled by the RET promoter to identify small molecules that inhibit the transcription of the RET gene. The RET gene plays a role in the development of medullary thyroid carcinoma (MTC) as part of multiple endocrine neoplasia type 2 (MEN2) syndrome. Adefovir dipivoxil was found to inhibit the transcription of the RET gene, reducing the expression of endogenous RET protein, inhibiting RET-dependent cell proliferation, and increasing apoptosis in MTC cells. Additionally, adefovir dipivoxil was found to interfere with Stat3 activity both in silico and in a biochemical assay, as well as in MTC cell lines. These results suggest that the cell-based assay is useful in identifying transcriptional inhibitors for oncogenes and has potential as a strategy to control cancer growth and development.

**Abstract:**

Aberrant gene expression is often linked to the progression of various cancers, making the targeting of oncogene transcriptional activation a potential strategy to control tumor growth and development. The RET proto-oncogene’s gain-of-function mutation is a major cause of medullary thyroid carcinoma (MTC), which is part of multiple endocrine neoplasia type 2 (MEN2) syndrome. In this study, we used a cell-based bioluminescence reporter system driven by the RET promoter to screen for small molecules that potentially suppress the RET gene transcription. We identified adefovir dipivoxil as a transcriptional inhibitor of the RET gene, which suppressed endogenous RET protein expression in MTC TT cells. Adefovir dipivoxil also interfered with STAT3 phosphorylation and showed high affinity to bind to STAT3. Additionally, it inhibited RET-dependent TT cell proliferation and increased apoptosis. These results demonstrate the potential of cell-based screening assays in identifying transcriptional inhibitors for other oncogenes.

## 1. Introduction

Medullary thyroid carcinoma (MTC) is an infrequent neoplasm comprising about 4% of all thyroid cancer cases. MTC is described as a tumor derived from the parafollicular C cells of the thyroid gland, which normally secretes calcitonin [1]. Approximately 75–80% of MTC cases are sporadic (non-inherited), while 20–25% are Familial (hereditary) [2]. MTC represents one of the most aggressive forms of thyroid cancer, with high rates of metastasis even in early disease stages [3,4]. The development of tumors in both sporadic and hereditary MTC is predominantly influenced by somatic and germline mutations in the RET proto-oncogene, respectively. [5]. RET encodes a single-pass transmembrane tyrosine kinase receptor on chromosome 10q11.2 [6]. The activation of RET mutations in MTC results in the constant activation of downstream signaling pathways, such as MEK/ERK and PI3K/AKT/mTOR, which subsequently promotes cell growth, invasiveness, and metastasis. [7,8]. As a result, RET has emerged as a crucial therapeutic target for MTC, especially through the use of tyrosine kinase inhibitor (TKI) therapy [9,10]. Evidence from past and ongoing preclinical and clinical research on the effectiveness of RET TKIs reinforces the idea that targeting mutant RETs is essential. Cabozantinib and vandetanib have demonstrated moderate success in treating metastatic MTC and initially improved cancer-specific survival durations [11,12,13,14]. Recently, the FDA approved selpercatinib (LOXO-292) for the treatment of MTC, as well as non-small cell lung cancer and advanced RET fusion-positive thyroid cancer. [15,16]. Resistance to TKI-based therapies is a significant issue in the treatment of MTC, especially metastatic MTC. [17,18]. TKIs are also known to induce a range of adverse effects in the majority of patients receiving treatment, such as hypertension, gastrointestinal disturbances, thyroid-related complications, fatigue, and weight loss [19]. Adverse events may require dose reduction, treatment interruption, or total discontinuation. Since current agents do not achieve optimal therapeutic effects, there is an urgent need to develop treatments that may overcome resistance and mitigate adverse effects to currently available multi-targeted TKIs. However, partly due to the escalating cost of new drug developments, alternative treatments are not available for cases of treatment resistant MTC. To overcome this lack of treatment options, we have explored drug repurposing as a strategy to discover alternative MTC therapies. The major advantage of this approach is that pharmacokinetic, pharmacodynamic, and toxicity profiles of candidate drugs have been already established in prior preclinical and Phase I studies [20,21,22]. Thus, drug repurposing may demonstrate improved efficacy, safety, and cost over standard de novo drug discovery, benefiting patients and drug developers.

To identify new drug candidates for MTC, we have explored novel RET, “rearranged during transfection” transcription inhibitors using a luciferase reporter-based cell assay. For high-throughput screening for novel RET transcriptional inhibitors, we utilized an HEK293 cell line stably expressing firefly luciferase gene under the control of a RET promoter as described in our previous studies [23]. Our previous studies have found this cell-based assay to be highly effective in drug screening aimed at discovering RET transcription inhibitors. We used the cell-based luciferase reporter assay in a 96-well format to screen an FDA-approved drug library from Selleckchem in this study. As described in this paper, adefovir dipivoxil was found to decrease RET-promoter-driven luciferase activity.

Adefovir dipivoxil, an oral diester prodrug, has been approved for the treatment of chronic hepatitis B (HBV) since 2002 [24,25]. The pivoxil moieties of adefovir dipivoxil enhances its cellular permeability and are promptly cleaved, both chemically and by esterase activity, to produce adefovir during absorption through the intestinal wall [24]. As an adenosine monophosphate analog, adefovir is readily phosphorylated into the active metabolite adefovir diphosphate by cellular kinases [26]. In the context of HBV treatment, this metabolite inhibits viral DNA polymerase by competing with the natural substrate dATP, causing chain-termination upon incorporation into viral DNA and exerting minimal effects on mammalian DNA polymerase [25,27]. The objective of this application was to evaluate the antitumor activity of the prodrug adefovir dipivoxil against MTC using robust preclinical models.

## 2. Materials and Methods

### 2.1. Cell Culture & Media

The human medullary thyroid carcinoma cell line (TT) was procured from the American Type Culture Collection (Manassas, VA, USA) and cultivated in DMEM 1640 medium (Cellgro, Manassas, VA, USA) enriched with 15% heat-inactivated fetal bovine serum (FBS). The isogenic cell line HEK293-RET, containing the luciferase reporter gene regulated by the wild-type RET promoter sequence, was produced as detailed in our prior research [23,28]. This cell line was cultured in DMEM 1640 media supplemented with 9% FBS. Dr. Rebecca Schweppe from the University of Colorado, Denver, supplied the TPC1 cells and the MZ-CRC-1 cells, which are derived from MTC. The TPC1 cells were cultured in RPMI-1640 medium with an added 9% of fetal bovine serum (FBS), while the MZ-CRC-1 cells were grown in DMEM/F-12 medium with a 15% supplementation of FBS. Primary Human Hepatocytes was purchased from Sigma Aldrich with its Human Hepatocyte Thawing Medium (MED-HTTM), Human Hepatocyte Culture Medium (MED-HHCM), Human Hepatocyte Plating Medium (MED-HHPMSP), and Supplement Pack (MED-HHCMSP). N-Thy-Ori 3-1 and K1 cell lines were grown in RPMI-1640 and Ham’s F12 medium (Thermo Fisher Scientific, Waltham, MA, USA) enriched with 9% FBS, respectively. All the cell lines were kept in a humidified environment containing 5% CO_2_ at 37 °C.

### 2.2. Cell Based Screening Using Luciferase Assay

The isogenic cell line HEK293-RET cell line was seeded in a 96-well plate at a density of 1.5 × 10^4^ cell perwell. Following this, the cells were subjected to treatment the subsequent days with the individual compounds from the Selleckchem FDA-approved library at a concentration of 5 μM and incubated up-to 48 h; the library was purchased from Sellecckehm (#L1300/100 μL/well). The cells were lysed with 25 μL passive lysis buffer (#E6110 Promega, Madison, WI, USA) and the luciferase expression was determined using the Steady-Glo Luciferase Assay System (Promega, Madison, WI, USA) following the manufacturer’s instruction. DMSO only was used as a negative control.

### 2.3. Reverse Transcription PCR Analysis of the RET mRNA Synthesis in TT Cells

The mRNA expression levels in TT and TPC1 cells following escalating concentrations of adefovir dipivoxil were assessed using RT-PCR as previously described [23]. The primers used for RT-PCR were as follows: RET forward: (5′-GCAGCATTGTTGGGGGACA-3′) and RET reverse (5′-CACCGGAAGAGGAGTAGCTG-30); Rpl9 forward (5′-CTGAAGGGACGCACAGTTAT-3′) and Rpl9 reverse (5′-ACGGTAGCCAGTTCCTTTCT-3′); RET/PTC1 forward (5′-GTCGGGGGGCATTGTCATCT-3′), and RET/PTC1 reverse (5′-AAGTTCTTCCGAGGGAATTC-3′). The PCR reactions involved an initial denaturation at 95 °C for 3 min followed by 33, 23 and 34 cycles for RET, Rpl9 and RET/PTC1, respectively, at 95 °C for 30 s, 52 °C for 30 s, and 72 °C for 30 s on a GeneAmp PCR system 9600 (Perkinelmer, Waltham, MA, USA). The PCR products were analyzed on 1.5% agarose gel electrophoresis.

### 2.4. Western Blotting

The whole-cell protein lysates were obtained by lysing cells using 2% CHAPS lysis buffer in the presence of 10 mM Tris-HCl, pH 7.4, 0/15 M NaCl, 5 mM EDTA, and Halt protease inhibitor cocktail (Thermo Fisher, Waltham, MA, USA). Then, the proteins were resolved on a 4–12% gradient polyacrylamide SDS-PAGE, which was purchased from Thermo Fisher, as described previously [28]. The primary antibodies used were as follows: anti-RET (#3220), mTOR (#2983), pmTOR (#5536), pSTAT3 (#9131), STAT3 (#4904), pChk1 (#2348),Chk1(#2360), H2AX (#9718), pAkt (#9271), and Akt (#9272), purchased from Cell signaling, Beverly, MA, USA, E2F1 (sc-56661); cyclin D (sc-20044), BCL-2 (sc-509) anti-pERK (sc-7383), anti-ERK (sc-271270), purchased from Santa Cruz Biotechnology; and anti-actin (ab8227) purchased from Abcam. Secondary antibodies consisting of mouse or rabbit IgG conjugated with horseradish peroxidase (HRP) were utilized (BioRad). For detection, an enhanced chemiluminescence kit from Santa Cruz Biotechnology was employed.

### 2.5. Cell Viability Assay

Cells were seeded at a density of 7500 cells/well in a 96 well plate and incubated overnight before being treated with adefovir dipivoxil across a broad range of concentrations for up to 96 h. The cell viability was assessed using 0.33 mg/mL MTS dye in combination with 200 µg/mL phenazine methosulfate (PMS) as described previously [23]. The absorbance was measured at 590 nm using a Synergy HT Multi-detection Microplate Reader (BioTek, Winooski, VT, USA).

### 2.6. Flow Cytometry

TT cells were subjected to cell-cycle analysis following the standard protocol as described before [23]. Cells were trypsinized, washed with ice cold PBS and fixed in 70% ethanol at −20 °C up to 24 h. The samples were then washed with ice-cold PBS and incubated in the presence of RNaseA (200 μg/mL) and propidium iodide (25 μg/mL) at 37 °C up to 30 min prior to analysis. The DNA fluorescence distributions were analyzed using a coulter Epics Profile flow cytometer and a cell-cycle histogram was generated for each sample.

### 2.7. Protein Preparation

The protein used for the docking study was the crystal structure of STAT3 with DNA complex (PDB: 1BG1). First, Schrödinger’s Protein Preparation Wizard tool was used to prepare the protein by removing the water molecules with less than 2 bonds to non-water, filling the missing side chains, refining, optimizing, and minimizing the protein structure using the OPLS4 force field. Once the prepared protein was generated, the optimized structure was utilized for further Grid generation.

### 2.8. Ligand Preparation

The 2D chemical structures of both Adefovir dipivoxil and Adefovir were converted into 3D and minimized using the Ligand Preparation tool in Schrödinger software. This was followed by a conformational search and generation of several 3D confirmations using the ConfGen application.

### 2.9. Ligand-Protein Molecular Docking and Prime MM-GBSA Calculations

The grid of the STAT3 crystal structure was generated using the Receptor Grid Generation tool and several amino acids were selected to determine the SH2 domain. These residues are Pro715, Phe716, Gln635, Lys591, Arg609, Ser611, and Ser613. After grid generation, several conformations were docked into the SH2 domain using Standard-Precision (SP) and Extra Precision (XP) scoring functions, followed by detailed analysis for the docked poses. Moreover, to enhance the accuracy of the docking results, Prime MM-GBSA was utilized to perform energy calculations for the docked poses, and VSGB was used as a solvation model and OPLS4 force field.

### 2.10. Biotinylation of 5′-Phosphate Group of Adefovir

Adefovir was first reacted with EDC (EDAC, 1-Ethyl-3-[3-dimethylaminopropyl]carbodiimide hydrochloride) and imidazole to create a highly reactive phosphorimidazolide. A resulting phosphorimidazolide was coupled to (+)-biotinamidohexanoic acid hydrazide to form a phosphoramidate bond with adefovir.

### 2.11. Spheroid Formation

Spheroids TT cells were created by plating about 7000 cells/well in ultra-low attachment (ULA) 96-well round-bottomed plates (Corning, NY, USA) as per the manufacturer’s protocol. TT cells were incubated with either vehicle or adefovir dipivoxil and photographed under an inverted microscope.

### 2.12. Xenograft Model

Male SCID (severe combined immunodeficiency) mice aged 8–10 weeks were used to investigate the effectiveness of adefovir dipivoxil treatment in vivo. The TT cell line was injected subcutaneously into the mice’s flank, and the tumor’s growth was closely monitored every three days by measuring and recording tumor diameter using vernier calipers. Tumors were measured three times a week during the treatment, and the volume of the tumors was assessed using the formula: tumor volume = (length × width^2^)/2, where length represents the largest tumor diameter and width represents the perpendicular tumor diameter. Once the tumor volume reached 100 mm^3^, the mice were randomly pair-matched to a treated group and a control group (*n* = 6/group). Adefovir was dissolved in 90% PBS and 10% DMSO and administered intraperitoneally as a single dose of 10 mg/kg for 5 days/week for 4 weeks. The relative tumor volume was calculated using the formula: relative tumor volume = Tx (absolute tumor volume on day X) × 100/(absolute tumor volume of the same tumor on day 0). The toxicity of the compound was assessed based on the average weight loss of the mice. All animal experiments were conducted in accordance with the Institutional Animal Care and Use Committee (IACUC) guidelines. The experiments were carried out in the Experimental Mouse Shared Resource (EMSR) animal facility laboratory (University of Arizona, Tucson, AZ, USA), which is accredited by the international Association for Assessment and Accreditation of Laboratory Animal Care (AAALAC).

### 2.13. Statistical Analysis

The results of at least three independent experiments are reported as mean ± SEM. To compare the means of multiple treatment groups and determine significant differences, statistical analyses were performed using either one-way ANOVA followed by Tukey’s post hoc test or two-way ANOVA followed by Bonferroni post hoc analysis. Student’s *t*-test was used to determine significant differences between two treatment groups where applicable. Differences between groups were considered significant at *p* < 0.05. Data analysis was performed using GraphPad Prism software (version 9.5.1-; GraphPad Software, Inc., San Diego, CA, USA).

## 3. Results

### 3.1. Adefovir Dipivoxil Is Identified as a Novel RET Transcription Inhibitor

We successfully explored the feasibility of using a luciferase reporter-based cell assay to identify FDA-approved small molecule drugs that would transcriptionally reduce RET expression. For high-throughput screening of novel RET transcriptional inhibitors, we previously generated an HEK293 cell line stably expressing firefly luciferase gene under the control of a 450-bp promoter sequence of the RET proto-oncogene (Figure 1A) [23]. In this study, HEK293-RET cells were treated with 5 µM concentration of the individual compounds, which were obtained from the Selleckchem FDA-approved drug library (>600 compounds) in 96-well plates and incubated up to 48-h. The effect of the test compounds on the RET promoter activity is determined by measuring the luminescence signal, which is generated from the luciferase expression. As shown in Figure 1B, most of the compounds had no or only a modest effect on luciferase gene expression, but four compounds (adefovir dipivoxil, viderabine, acyclovir, and thioguannine) showed a significant decrease (>75%) in the luminescence signal. These four compounds were selected as initial candidates to further validate their effects on luciferase expression in an independent experiment at increasing drug concentrations. Among the four lead compounds, only adefovir dipivoxil (Figure 1C) showed a dose-dependent decrease in luciferase activity (Figure 1D), which prompted further investigation in the study.

### 3.2. Adfovivir Dipivoxil Affects RET Expression and RET Signaling Pathway

Next, we measured protein and mRNA expression for RET gene in MTC TT cells treated with adefovir dipivoxil. As shown in Figure 2, adefovir dipivoxil reduced RET protein (Figure 2A) and mRNA expression (Figure 2B) in TT cells in a dose dependent manner after 24 and 48 h treatment. To determine whether the transcriptional repressive effect of adefovir dipivoxil in MTC TT cells is specific to the RET promoter region, we used the papillary thyroid carcinoma (PTC) TPC1 cell line. In PTC TPC1 cells, the tyrosine kinase domain of RET is fused with the 5′-region of CCD6 gene due to chromosomal rearrangement, which results in a chimeric RET/PTC1 fusion protein [28]. Hence, the transcriptional activation of the RET/PTC1 gene in TPC1 cells is regulated by the CCD6 gene promoter region. Even at a high concentration of adefovir dipivoxil (5 µM) for 48 h, the expressions of RET/PTC1 mRNA and protein were not reduced in TPC1 cells, implying that the inhibitory effect of adefovir dipivoxil may act through the RET promoter region (Figure 2C). Moreover, the level of the RET protein in the MZ-Crc-1 cell line, which harbors a mutated RET gene (M918T), did not alter after being treated to different concentrations of adefovir dipivoxil for 48 h, as shown in Appendix A. One limitation of this study is that RT-PCR was utilized instead of a more quantitative assay, qRT-PCR.

Numerous studies have shown that the PI3K/AKT/mTOR pathway is an essential signaling pathway that is activated by the oncogenic RET, which subsequently promotes the survival of cancer cells through the phosphorylation of mammalian target of rapamycin (mTOR) protein [7]. Hence, we evaluated the effect of adefovir dipivoxil on the PI3K/Akt/mTOR pathway in TT. As shown in Figure 2D, adefovir dipivoxil reduced the levels of pmTOR, pAkt, and pERK after 48 h of treatment. This suggests that the reduction in the expression of downstream targets such as pmTOR, pAkt, and pERK by adefovir dipivoxil is due to its targeting of RET expression in TT cells.

### 3.3. Effects of Adefovir Dipivoxil in Cell Proliferation

To confirm the selectivity of adefovir dipivoxil for TT cells over normal human cells, we used primary human thyroid (N-Thyi-Ori 3-1) and hepatocytes as control cell lines. We assessed cell growth inhibition using the MTT assay by culturing each cell line in various concentrations of adefovir dipivoxil for 96 h. As shown in Figure 3, the TT cell line exhibited a significant dose-dependent suppression of growth in response to adefovir dipivoxil, with an IC50 of 0.62 µM. We also evaluated the effect of adefovir dipivoxil in the Mz-CRC-1 cell line, a model of MTC with a mutation in Met918 → Thr in exon 16 of the RET gene, which showed a less pronounced inhibitory effect compared to TT cells. Furthermore, we tested the effect of adefovir dipivoxil in two different papillary thyroid carcinomas (PTC): TPC1 and K1. TPC1 has RET/PTC1 fusion, while K1 has a wild-type RET [29]. Adefovir dipivoxil was less effective in inhibiting the growth of both papillary thyroid cancer cell lines. The concentrations of adefovir dipivoxil required to suppress the growth of non-TT cell lines were at least 10-fold higher than those required to inhibit TT cell growth (Figure 3).

### 3.4. The Effects of Adefovir on Egr-1 and CREB

In this study, we aimed to further understand the molecular mechanisms underlying the effects of adefovir dipivoxil on RET expression in TT cells. Since previous study suggested that Egr-1 induction plays a key role in RET gene expression by directly binding to the RET minimal promoter, we first examined the effects of adefovir dipivoxil on Egr-1 expression in TT cells [30,31,32]. As shown in Figure 4, adefovir dipivoxil significantly increased the expression of Egr-1, suggesting that the downregulation of RET expression in TT cells is at least partly linked to the induction of Egr-1. The CREB pathway is known to participate in Egr-1 induction by interacting with Egr-1 promoter [33]. Thus, we also examined the effect of adefovir dipivoxil on the CREB pathway. In agreement with the proposed link between Egr-1 induction and CREB pathway, we observed that both CREB and its activated form pCREB-Ser133 (pCREB) are increased in TT cells treated with adefovir dipivoxil. These observations suggested that pCREB may play a role in regulating adefovir dipivoxil-mediated Egr-1 expression, resulting in the downregulation of RET expression in TT cells.

### 3.5. STAT3 as a Potential Cellular Target for Adefovir

The signal transducer and activation of transcription 3 (STAT3) is activated through phosphorylation by JAK1/2 kinase [34]. Previous research suggests that RET protein can also phosphorylate STAT3 through its intrinsic kinase activity in a JAK1/2 independent manner, specifically at the Y705 and S727 residues, which are necessary for optimal transcriptional activity [35,36]. Upon activation, STAT3 homodimerizes or heterodimerizes with other members of the STAT family, facilitating nuclear translocation. Once inside the nucleus, STAT3 binds to its promoter-specific region to stimulate the expression of cell-cycle regulatory genes, thereby promoting cell proliferation and transformation [37]. In the present study, we hypothesized that STAT3 may be a potential cellular target for adefovir dipivoxil. Adefovir dipivoxil is structurally similar to fludarabine phosphate, a nonselective inhibitor of the signal transducer and an activator of transcription proteins (STATs) [34]. A docking study was performed for Adefovir dipivoxil and Adefovir to see if the two forms would possess a distinct interaction or binding mode at the STAT3 SH2 domain binding site. The docking study was performed using the Glide tool in Maestro Schrödinger software, and two scoring functions were utilized to perform a quick scoring rank for the drug-protein complex, followed by more accurate and rigorous Prime MM/GBSA binding free energy calculations for the docked poses. Our SP and XP docking results showed that Adefovir had slightly higher docking scores relative to Adefovir dipivoxil; however, this was not the case with the Glide energy and MMGBSA binding free energy that demonstrated higher energies for Adefovir dipivoxil (results are in Table 1). The 2D interaction diagram for Adefovir dipivoxil showed hydrogen bond interactions with Arg609, Ser613, Ser636, Lys591, and a pi-cation interaction was formed with Lys591 residue of STAT3. As illustrated in Figure 5, the binding mode of adefovir dipivoxil demonstrated an extension of one of the ester arms to an extra binding pocket that is not occupied by Adefovir which could contribute to the difference in the experimental inhibition of STAT3 phosphorylation by the two drugs (Appendix A).

To further validate STAT3 as a potential cellular target of adefovir dipivoxil, small scale affinity columns using adefovir dipivoxil immobilized to an agarose gel were used for capturing target proteins. Eluted proteins were separated by SDS-PAGE, and western blot analysis was used to detect STAT3 protein in the eluted sample (Figure 6). The result of our pull-down assay further confirms the existence of the adefovir dipivoxil–STAT3 interaction as predicted by our modeling study.

### 3.6. Adefovir Dipivoxil Effects on STAT3 and pSTAT3 in MTC Cell Line

After identifying STAT3 as a potential target for adefovir, the next step was to determine if adefovir dipivoxil would impact the phosphorylation of Tyr705 in STAT3 in TT cells. The phosphorylation of Tyr705 in STAT3 is a known result of cytokine stimulation in cells. The results, shown in Figure 7, demonstrate that after extended exposure (24 and 48 h) to adefovir dipivoxil, there was a significant decrease in the phosphorylation of Tyr705 in STAT3. This supports the idea that STAT3 is a target for adefovir dipivoxil.

### 3.7. Cell Cycle Effects of Adefovir Dipivoxil

The PI3K/Akt/mTOR pathway is known to stimulate cancer cell proliferation by mediating the transition from G1 to S phase during cell cycle progression through the activation of a cell-cycle regulator protein, cyclin D1 [38,39,40]. As shown in Figure 8A, adefovir dipivoxil decreased cyclin D1 protein expression in TT cells, which was determined by western blotting after 48-h treatment. The decline in cyclin D was accompanied by an elevation in the regulatory cell cycle protein p53 and a reduction in the anti-apoptotic protein BCL2. To further validate that the reduced levels of cyclin D1 inhibits cell-cycle progression, flow-cytometry analyses was performed in these cells in the absence and presence of adefovir dipivoxil (5 µM) after 48 h of exposure. As shown in Figure 8B, treatment of TT cells with adefovir dipivoxil caused an accumulation of cells in S-phase within 48 h, as well as increasing percentages of cells in a sub-G1 peak. This result suggests that, on cellular entry, adefovir dipivoxil can be phosphorylated to a triphosphate form and incorporated into nuclear DNA, which is critical for its toxicity. Incorporation can cause steric hindrance of extending replication forks, which induces S-phase arrest and apoptosis.

### 3.8. Characterizing DNA Damage Checkpoints Activated by Adefovir

The incorporation of adefovir into nuclear DNA can cause steric hindrance of extending replication forks, which induces S-phase arrest [41]. In response to DNA damage, ATR phosphorylates and activates the protein kinases Chk1. Thus, Chk1 is activated following DNA damage in the S phase [42]. To examine the involvement of checkpoint kinases in adefovir-induced DNA damage, we examined Chk1 phosphorylation by immunoblotting as previously described [43]. As shown in Figure 9A, adefovir dipivoxil increases the levels of phosphorylated H2AX (γ-H2AX), which is a sensitive marker for DNA damage. The activation of Chk1 is known to induce E2F1 protein in response to DNA damage [42]. The TT cells treated with various concentrations of adefovir dipivoxil showed increased phosphorylation of Chk1 at Ser345 and expression of E2F1. Overall, our results confirm adefovir dipivoxil induced DNA damage in TT cells. We also examined whether the activation of the DNA damage checkpoint is associated with RET downregulation. However, treatment of TT cells with adefovir dipivoxil in the presence of the Chk1 inhibitor PF-477736, which is known to abrogate the DNA damage checkpoint, showed little effect on the downregulation of RET (Figure 9B). This result suggests that the downregulation of RET by adefovir dipivoxil is dissociated from the activation of the DNA damage checkpoint.

### 3.9. Signaling Pathways Involved in the Migration, Invasion, and Resistance of TT Cells and Spheroid Formation

Active migration of tumor cells is essential for the invasion of and metastasis to internal organs [44]. Thus, in this study, the effect of adefovir dipivoxil on migration was evaluated in using a wound healing assay according to a published procedure [45]. As shown in Figure 10A, adefovir dipivoxil significantly diminishes cell migration compared to the respective control groups.

During the metastatic process, cancer cells exhibiting high proliferation and relatively low invasiveness undergo a transition into highly invasive mesenchymal cells via epithelial-mesenchymal transition (EMT) [46]. This transformation entails a loss of epithelial features and acquisition of mesenchymal attributes, facilitating cell invasiveness. Pleiotropic transcription factors such as Snail1, Slug, Twist1, and ZEB1 govern these morphological and molecular alterations, with their expression commonly observed during EMT, resulting in invasion, dissemination, and metastasis [47,48,49]. A concomitant increase in N-cadherin and decrease in E-cadherin represents a key hallmark of metastatic tumor cell (MTC) metastasis [50]. As illustrated in Figure 10B, adefovir dipivoxil exhibits dose-dependent suppression of mesenchymal marker expression, encompassing ZEB1, vimentin, Slug, and Snail. This finding highlights the inhibitory impact of adefovir dipivoxil on epithelial-mesenchymal transition (EMT) in TT cells.

We investigated the influence of adefovir dipivoxil on the three dimensions of growth of TT spheroids in non-adherent, serum-free culture conditions. Adefovir dipivoxil notably diminished the spheroid size over time, with cell death progressing from the periphery inwards to the spheroid core [51] (Figure 11).

### 3.10. In Vivo Antitumor Effect of Adefovir Dipivoxil in Xenograft Models of Human Medullary Thyroid Cancer

In the final phase of our current study, we assessed the in vivo activity of adefovir dipivoxil on the tumor growth of the MTC xenograft model. SCID mice were injected with TT cells subcutaneously and tumor growth was assessed using a vernier caliper. As shown in Figure 12, tumor growth inhibition was observed in mice that received 10 mg/kg adefovir dipivoxil compared to the vehicle group. After 45 days, the survival rate of mice receiving adefovir dipivoxil was 80%, compared to 40% in the untreated group (Figure 12B).

## 4. Discussion

High-dose therapy with adefovir dipivoxil has been associated with a form of reversible nephrotoxicity called proximal renal tubular dysfunction (PRTD), which primarily targets proximal tubules [52,53]. However, PRTD most commonly occurs after six or more months of daily oral administration of 120 mg adefovir dipivoxil, with 5.8% of treated patients developing nephrotoxicity by six months versus 0.4% for a placebo group [54]. As such, short-term treatment (about three weeks) may not produce noticeable renal dysfunction or toxicity. Since most FDA-approved chemotherapeutic drugs for MTC have more serious side effects [10], adefovir dipivoxil could offer significant advantages over existing drugs, especially if nephrotoxic effects are mitigated.

The main objective of the present study is to generate new therapeutic options for metastatic MTC, which may complement or replace existing chemotherapies to extend the lifespan of patients with MTC and improve their quality of life. It is also innovative to repurpose FDA-approved non-oncologic drugs as potent anticancer agents for MTC patients, due to the advantages of repurposing over de novo drug development (e.g., bypassing the need for additional phase I safety and dosing clinical trials) [20]. Although FDA-approved non-oncologic drugs have been investigated as potential antitumor agents for other types of cancer, adefovir dipivoxil is the first ever FDA-approved nononcologic drug to show selectively cytotoxic effects on MTC.

In this study, we investigated how adefovir dipivoxil reduces the proliferation of TT cells by downregulating RET expression. Oncogenic RET activation stimulates cell proliferation and survival by triggering multiple intracellular signaling cascades. In an earlier study, Drosten et al. characterized the downstream signaling pathways associated with RET required for tumor progression and maintenance in TT cells [55,56,57,58]. They disrupted RET phosphorylation and activation in TT cells by using an adenoviral vector expressing the truncated dominant-negative RET protein, which lacks the intracellular tyrosine kinase domain, leading to the suppression of the Raf/MEK/ERK and PI3K/Akt/mTOR pathways, implying their primary role in RET-mediated transformation. In alignment with this research, our findings revealed that the inhibition of RET expression in TT by adefovir dipivoxil diminished the phosphorylation of ERK, AKT, and mTOR, and subsequently reduced expression of cyclin D1, P53, and Bcl-2, which are regulated by the Raf/MEK/ERK and PI3K/Akt/mTOR pathways (Figure 8A). Although MZ-CRC1, an MTC cell line with the M918T mutation, appears to be more resistant to adefovir dipivoxil treatment compared to TT cells carrying the C634W mutation (Appendix A), it can be hypothesized that specific genes might be highly overexpressed in the RET M918T mutant but not in the C634W mutant. This overexpression could contribute to the increased resistance of MZ-CRC1 to the treatment [11,28,29,30,31,32,33,34,35,36,37,38,39,40,41,42,43,44,45,46,47,48,49,50,51,52,53,54,55,56,57,58]. Notably, we also found that normal thyroid cells, Nthy-ori-3-1, and hepatocytes cell lines showed significant resistance to adefovir dipivoxil, indicating that this compound is more selective for MTC.

Cancer metastasis, the spread of cancer cells to distant organs within the body resulting in new tumor formation, is the primary cause of cancer-related death in MTC [11,56,57,58,59]. Metastasis begins with the detachment of cancer cell(s) from the primary cancer site, which are spread to a secondary site by the vascular and/or lymphatic system where the cancer cell(s) continue to grow and proliferate [60]. During metastasis, activation or repression of various genes occurs. Recent studies have shown that the overexpression of Egr-1 in breast cancer can suppress cell migration and invasion. The aforementioned inhibition was augmented by the induction of CREB and NM23-H1, respectively [59].

Egr-1, a transcription factor, has a critical function in processes such as tissue injury, immune responses, and fibrosis. In addition, recent research has highlighted its significance in the development and progression of cancer. Egr-1 is believed to have a significant impact on tumor cell proliferation, invasion, metastasis, and angiogenesis. However, the precise mechanisms by which Egr-1 regulates these processes are not yet fully understood [61]. A low level of Egr-1 and CTCF is usually observed in invasive breast cancer. Hence, Egr-1 overexpression has been found to suppress tumor growth in several forms of cancer [62]. In MTC, some studies have linked the overexpression of Egr-1 directly to RET downregulation [31,63,64]. This effect may be related to the ability of Egr-1 to remove other transactivators such as SP1 and SP3 from the G-rich region in RET promotor [33,34]. Some speculative studies also suggest that the expression of RET is maintained at a low level by the presence of Egr-1 [64,65]. In our study, adefovir dipivoxil increased expression of Egr-1, resulting in a significant shift in the ratio of Egr-1 to RET in vitro. This shift was accompanied by an increase in the Ser-133 phosphorylation of CREB, which is essential for Egr-1 transcription. Recent studies have revealed that Egr-1 is capable of modulating crucial proteins involved in the cell cycle, including p53 and Cyclin D [65]. Additionally, Egr-1 has been found to influence the expression of BAX and BCL2 in pancreatic cancer cells, with an increase in BAX and a decrease in BCL2 observed in multiple experimental conditions [66]. These findings were consistent with the results of multiple testing conditions in MTC model.

STAT3 activation has been consistently observed in a high percentage of cancerous cell lines, animal models, and human tumors [67]. This activation is typically linked to a poorer prognosis as it contributes to uncontrolled cell growth and alteration of the tumor’s surrounding environment. Consequently, targeting STAT3 is viewed as a promising approach for treating cancer. Although no drugs have demonstrated significant efficacy in clinical trials, it has been suggested that combining STAT3 inhibition with other treatments, such as targeted therapies or radiotherapy, could lead to improved clinical outcomes [68]. In MTC, the constitutive activation of RET proto-oncogene results in trans-phosphorylation of several tyrosine residues [69]. These residues recruit and interact with other proteins to activate the pivotal signaling pathway, which is essential for cell survival and differentiation. One such activated protein is STAT3. RET has been reported to have two STAT3 docking sites, Y752 and Y928 [70,71]. Transactivation of STAT3 is essential for cell tumorigenesis and is independent of the JAK and Src pathways [72]. This activation is a result of the phosphorylation of both Tyr705 and Ser727 of STAT 3 by RET [72]. Canonical STAT3 activation involves phosphorylation of Y705, resulting in STAT3 dimerization, nuclear translocation, and binding to a DNA. Due to the dimerization process of STAT3 proteins, many small peptides or peptidomimetics, as well as small molecule inhibitors, have been created. These molecules target the interaction between the SH2 domain and the phosphorylated Y705 residue [73].

A recent study has reported that there are three binding “hot spots” identified in the STAT3 SH2 domain [35,74,75]. Among these “hot spots” is the pTyr705 binding site, which includes Lys591, Arg609, Ser611, Glu612 and Ser613 [72,73,74,75]. Our in silico analysis has shown that adefovir dipivoxil binds to the STAT3 SH2 domain with high affinity. We also have examined the effect of both adefovir and adefovir dipivoxil in vitro. Our in vitro study shows that adefovir dipivoxil was able to pull down STAT3. We also showed the effect of adefovir dipivoxil on Tyr705 phosphorylation status in TT MTC cells. The result of the docking simulations of adefovir dipivoxil to the STAT3 SH2 domain, when combined with the biochemical analysis of STAT3, STAT3 phosphorylation, and STAT3 downstream proteins, provides compelling evidence of the direct binding and effect of adefovir dipivoxil on STAT3.

In addition to the activation of RET and STAT3, it has been noted that CDK5 regulates STAT3 in medullary thyroid cancer, influencing cell proliferation [36]. Suppressing both STAT3 and CDK5 has been shown to slow down human medullary thyroid cancer cell proliferation [36]. Furthermore, overexpression of a mutant STAT3 (Ser 727 to Ala) has a negative impact on the CDK5 pathway and inhibits TT cell growth, highlighting the significant role of STAT3 Ser 727 and the CDK5 pathway in MTC treatment [36]. The roles of CDK5 and STAT3 were not extensively explored in this study. We believe that further investigation of these roles is necessary for a deeper understanding of the interplay between STAT3, RET, and the CDK5 pathway.

ATR and Chk1 play an essential role in cell cycle progression, replication, and homologous recombination repair [39,40,41]. Deletion of ATR and Chk1 has been reported to cause lethality in mice, demonstrating the pivotal role of the ATR pathway in early embryonic development [76]. The ATR pathway is required to prevent DNA integrity loss in the S phase [77]. Chk1 is downstream of the ATR pathway. During the activation of the ATR pathway, Ser 317 and Ser 345 in Chk1 are phosphorylated by ATR. Chk1 later auto-phosphorylates ser 296 and inhibits CDC 25 [41,77]. Chk1 regulates the activity of cyclin-dependent kinases and cell cycle progression. In our study, we demonstrated activation of the ATR pathway after Adefovir dipivoxil treatment in the MTC model. This activation is associated with increased phosphorylation of Chk1 in TT cells. Chk1 activation also led to the induction of E2F. Moreover, the presence of irreparable double-stranded breaks was observed due to the accumulation of pγH2AX. Thus, we observed S phase arrest of TT cells after adefovir dipivoxil treatment. The effect of adefovir dipivoxil on TT cells may be related to the combination of inhibiting the RET signaling pathway and ATR activation. This could explain the diverse effects of adefovir on other cell types that are not driven by RET-related cancer, RET/PTC1, and RET M918T. Additional studies are needed to gain a deeper understanding of the mechanisms involved in ATR-mediated DNA damage responses and how they differ among distinct cancer cells [78,79]

Epithelial-mesenchymal transition (EMT) is a dynamic process that controls cellular phenotype and function [80]. When cells undergo EMT, epithelial cells lose their cell-cell contact, adhesion, and polarity. In MTC, the EMT process allows tumor cells to migrate from the primary site to distant sites such as bone, lymph nodes, and liver. In MTC, RET constitutive activation accelerates EMT, which increases aggressiveness of the tumor [81,82,83]. The constant activation of RET increases cell migration by inducing lamellipodia formation [83]. In the EMT process, the AKT pathway plays a pivotal role in controlling actin dynamic [84]. The inhibition of PI3k after adefovir treatment downregulates the downstream AKT and mTOR proteins. This inhibition prevents invasion characteristics of TT cells in vitro. In our study, the cell migration inhibition was accompanied by the downregulation of a major EMT proteins. This effect is a result of RET downregulation in the MTC cell line. Additionally, Egr-1 has been shown to have a role in inhibiting tumor invasion and metastasis across various types of cancers [85]. In non-small cell lung cancer, TGF-b1 has been found to decrease Egr-1-induced EMT of cancer cells, and high Egr-1 expression has been correlated with reduced EMT [85]. This effect is possibly mediated by Egr-1′s ability to regulate the expression of SNAIL, SLUG, and E-cadherin. Oxytocin has been demonstrated to increase Egr-1 expression via an EGFR- and ERK-dependent pathway in head and neck squamous cell carcinoma, which may inhibit tumor invasion and metastasis by promoting E-cadherin overexpression [86]. In hepatocellular carcinoma, Egr-1 expression is induced by b-lapachone, which may hinder invasion and metastasis by impacting TSP1, SNAIL, and E-cadherin expression [87]. Thalidomide has been found to upregulate Egr-1 and inhibit the metastasis of leukemic cells [88]. Moreover, LY294002 has been shown to impede the invasiveness and metastasis of leukemic cells by increasing Egr-1 expression via a mechanism not reliant on the PI3K-AKT pathway. [89]. In this study, the impact of Adefovir on several proteins essential for epithelial-mesenchymal transition (EMT) has been demonstrated, including Snail, Slug, N-cadherin, E-cadherin, and Vimentin.

After all, our study showed that adefovir dipivoxil at a low dose of 10 mg/kg can successfully reduce MTC tumor growth when given systemically, without causing any significant toxicity. This finding is important in highlighting the therapeutic potential of adefovir dipivoxil. Further optimization of its efficacy may be possible by increasing the concentration of adefovir or using combination therapy instead of monotherapy.

## 5. Conclusions

In summary, the study suggests that adefovir dipivoxil holds promise as a potential therapy for metastatic medullary thyroid carcinoma (MTC). The drug’s ability to induce cytotoxic effects in MTC cells and its approval by the FDA make it a more favorable option compared to existing chemotherapies that have more severe side effects. The study also sheds light on the molecular mechanisms involved in MTC metastasis, highlighting the crucial roles of RET, Egr-1 and STAT3 in tumor proliferation, invasion, and angiogenesis. These findings provide valuable insights for developing new strategies to inhibit these pathways and improve patient outcomes. The existing knowledge of adefovir dipivoxil’s side effect profile can facilitate its further development as a potential therapeutic agent for MTC. Although adefovir dipivoxil treatment is associated with mild and reversible nephrotoxicity, this side effect is only observed in a few patients treated with high doses of the drug for over six months. The drug repurposing approach can leverage pre-existing clinical data to reduce the amount of time and work needed for further development. Adefovir dipivoxil’s potential to be repurposed as a new therapeutic option for MTC patients who do not respond to current treatments or lack other therapeutic options is significant. The entire process of drug development from design to clinical trials typically takes around 10 to 12 years, but drug repurposing can make the process safer, faster, and more cost-effective.

## Figures and Tables

**Figure 1 cancers-15-02163-f001:**
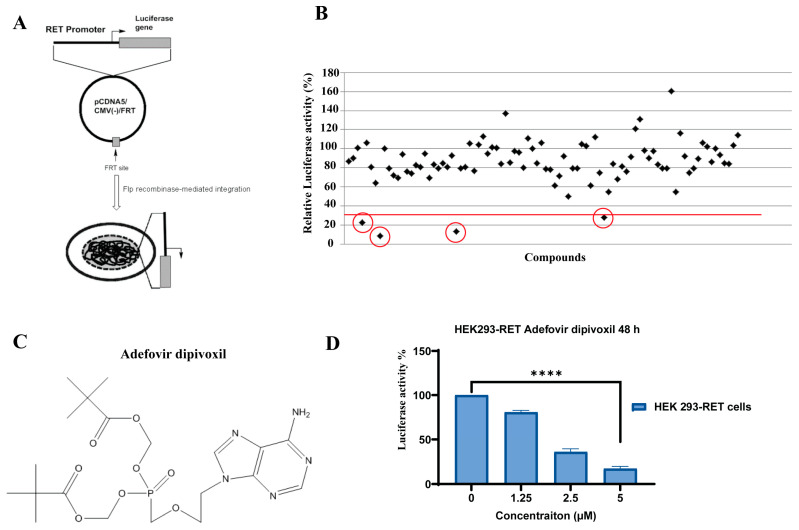
The cell-based screening of the Sellechem FDA approved drug library using the HEK293-RET cells involves the identification of potential drug candidates. (**A**) Scheme for the generation of Flp-In cell line HEK293-RET cell line using the Flp-In expression vector pCDNA5/CMV(-)/FRT/RET; (**B**) Right panel effects of individual compounds (5μM) from a single 96-well plate on the luciferase activity in HEK293-RET cells to screen for a potential transcriptional repressor for RET gene. (**C**) Structure of adefovir dipivoxil. (**D**) Validation of the initial hits from the screening assay by performing individual experiment at various concentrations of adefovir dipivoxil up to 48 h. Luciferase activity was measured as relative luminescence units (RLU) and normalized to total protein content. Experiments were performed in triplicate with normalized quantification and analysis of the difference between concentration by Student’s *t*-test (*n* = 3; **** *p* ≤ 0.0001).

**Figure 2 cancers-15-02163-f002:**
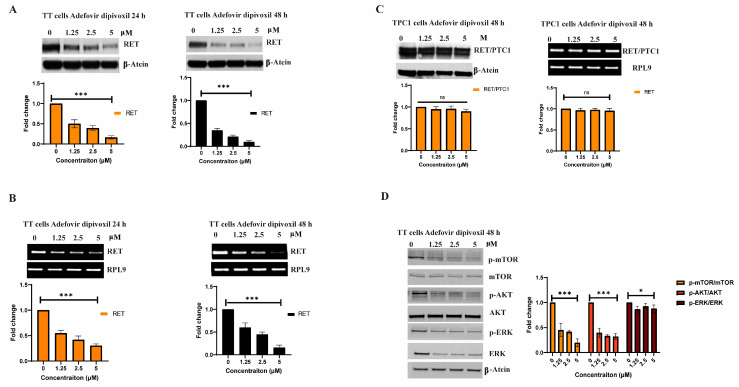
Effects of adefovir dipivoxil on RET expression in TT cell line. (**A**) Effect of adefovir dipivoxil on the RET expression in TT cells was determined by western blotting following 24 and 48-h incubation (**B**) Effect of adefovir dipivoxil on the RET mRNA expression in TT cells after 24 and 48-h treatment at various concentrations. (**C**) Effect of adefovir dipivoxil on the RET protein expression (left panel) and mRNA expression (right panel) after 48 h treatment at different concentrations. (**D**) Cellular effects mediated by RET down-regulation proteins including p-mTOR, mTOR, pAKT, AKT, pERK and ERK after 48 adefovir dipivoxil in TT cells were determined by western blotting. The western blotting results were quantified by densitometric analysis with normalization to basal expression (total protein) and differences among groups were analyzed by one-way ANOVA with Tukey’s post hoc test (*n* = 3; ns, not significant; * *p* ≤ 0.05; *** *p* ≤ 0.001). The uncropped blots are shown in Appendix A.

**Figure 3 cancers-15-02163-f003:**
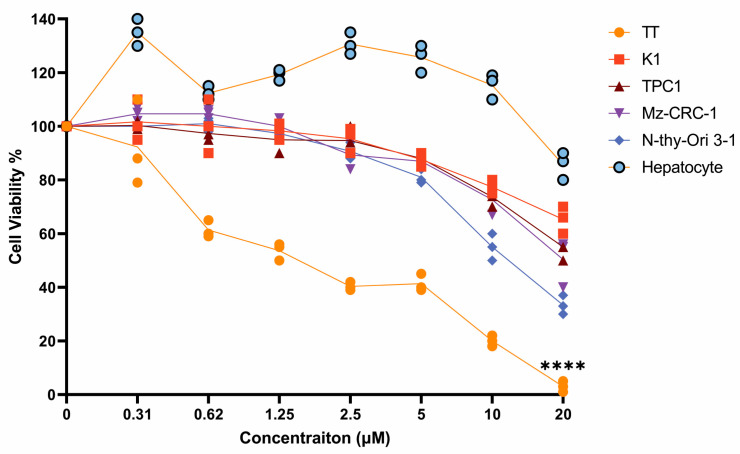
Adefovir dipivoxil effects on cell viability. MTS assay for the TT, K1, TPC1, Mz-CRC-1, Nthy-Ori 3-1, and Hepatocyte cell lines, treated with an increasing concentration of adefovir-dipivoxil for 96 h to determine the cell viability. Data are mean +/− SEM of three different experiments and the differences among groups were analyzed by one-way ANOVA with Tukey’s post hoc test (**** *p* ≤ 0.0001).

**Figure 4 cancers-15-02163-f004:**
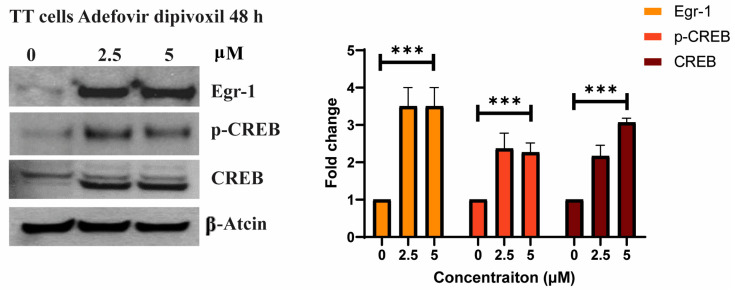
The effects of adefovir dipivoxil on the Egr-1 expression and CREP pathways in TT cells. The levels of Egr-1, CREB, and pCREB were determined by western blotting following 48-h incubation of TT cells adefovir dipivoxil. The western blotting results for Egr-1, p-CREB, and CREB were quantified by densitometric analysis with normalization to basal expression, and differences among groups were analyzed using one-way ANOVA (*n* = 3; *** *p* ≤ 0.001). The uncropped blots are shown in Appendix A.

**Figure 5 cancers-15-02163-f005:**
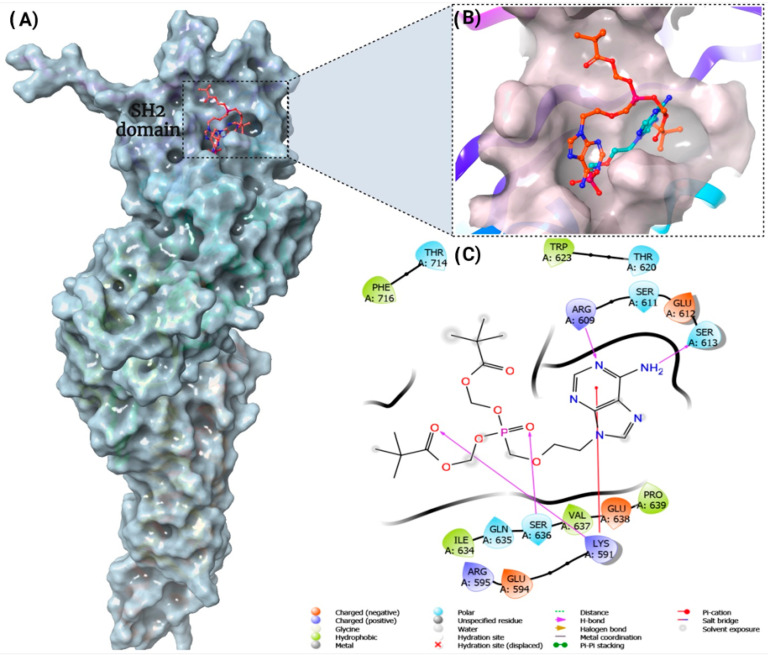
Molecular docking of adefovir dipivoxil and adefovir against STAT3 SH2 domain. (**A**) Docking of Adefovir dipivoxil in STAT3 SH2 binding pocket. (**B**) Overlay of Adefovir dipivoxil and Adefovir Docked Poses in the STAT3 SH2 domain. (**C**) The 2D interaction map of the XP docking pose of Adefovir dipivoxil in STAT3 SH2 binding pocket. Adefovir dipivoxil showed hydrogen bond interactions with Arg609, Ser613, Ser636, Lys591, and a pi-cation interaction was formed with Lys591 residue of SH2 domain of STAT3.

**Figure 6 cancers-15-02163-f006:**
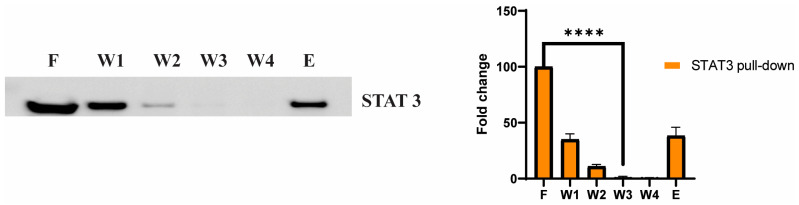
Pull-down assay: STAT3 capture from TT cell extracts with adefovir dipivoxil affinity agarose by pull-down assays. Supernatant (F) corresponds to the cell extract after incubation with each type of affinity bead, which was used as a control of the input material, and the gels were washed four times (W1, W2, W3 and W4). Bound STAT3 (E) was eluted with SDS-PAGE protein loading buffer. Experiments were performed in triplicate with normalized quantification and analysis of the difference between each step (F, W1, W2, W3, W4 and E) by Student’s *t*-test (*n* = 3; **** *p* ≤ 0.0001). The uncropped blots are shown in Appendix A.

**Figure 7 cancers-15-02163-f007:**
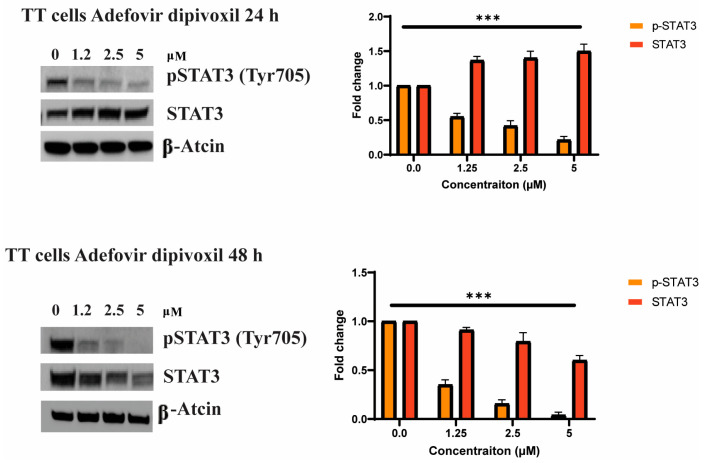
Inhibition of STAT3 tyrosine phosphorylation in TT cells by adefovir dipivoxil. TT cells were treated with various concentrations of adefovir dipivoxil at 24- and 48 h. Cell lysates were subjected to Western blot analysis with anti-phospho-Tyr705 STAT3 (pTyr-STAT3) or anti-STAT3 (STAT3) as indicated. The western blotting results for pSTAT3 and STAT3 were quantified by densitometric analysis with normalization to basal expression, and differences among groups were analyzed using one-way ANOVA (*n* = 3; *** *p* ≤ 0.001). The uncropped blots are shown in Appendix A.

**Figure 8 cancers-15-02163-f008:**
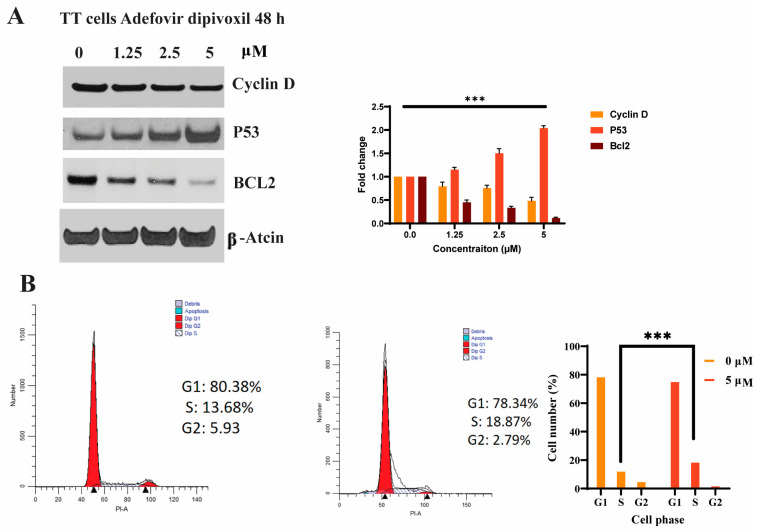
Cell cycle analysis for TT cells after adefovir dipivoxil treatment. (**A**) Expression of Cyclin D, P53, and Bcl2 in TT cells after 48 h adefovir dipivoxil treatment. The western blotting results for Cyclin D, P53 and BCL2 were quantified by densitometric analysis with normalization to basal expression, and differences among groups were analyzed using one-way ANOVA (*n* = 3; *** *p* ≤ 0.001). (**B**) Flow cytometry cell cycle analysis using propidium iodide DNA staining in untreated and adefovir dipivoxil-treated cells. The cell cycle data are mean +/− SEM of three different experiments and the differences among groups were analyzed by one-way ANOVA with Tukey’s post hoc test (*** *p* ≤ 0.001). The uncropped blots are shown in Appendix A.

**Figure 9 cancers-15-02163-f009:**
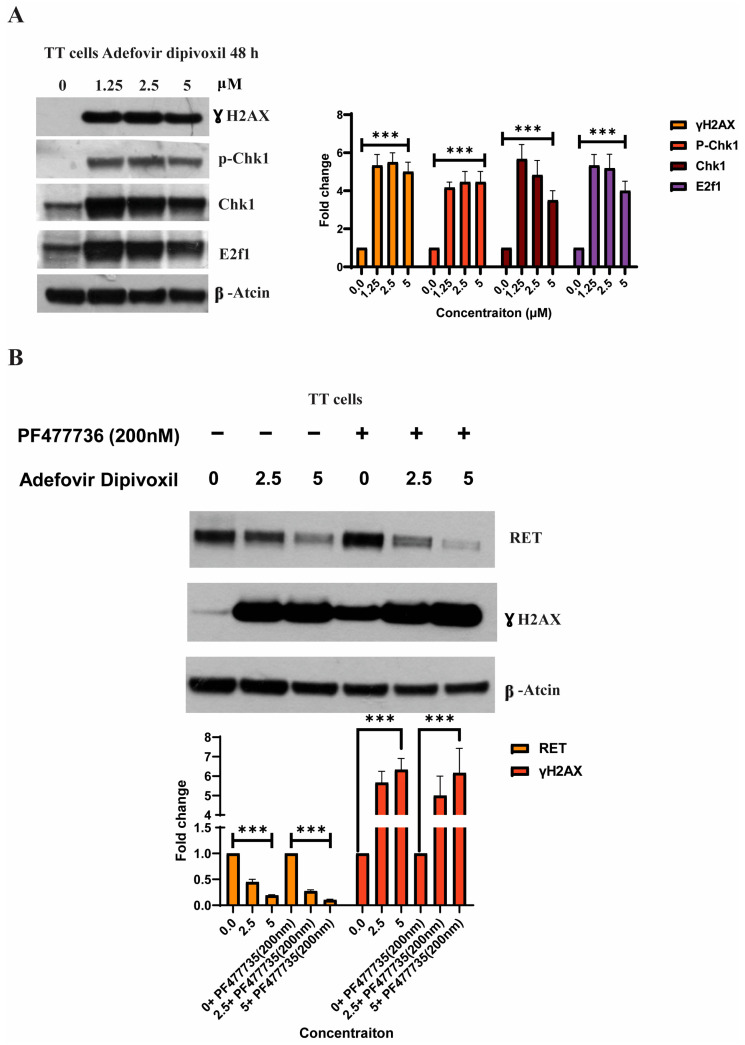
Evaluating DNA damage in TT cells after adefovir treatment. (**A**) Activation of DNA damage checkpoint by adefovir dipivoxil in TT cells. The western blotting results for γ-H2AX, p-Chk1, Chk1, and E2f were quantified by densitometric analysis with normalization to basal expression, and differences among groups were analyzed using one-way ANOVA (*n* = 3; *** *p* ≤ 0.001) (**B**) and effect of Chk1 inhibitor PF-477736 on adefovir dipivoxil-induced RET downregulation in TT cells. The western blotting results for RET and BCL2 were quantified by densitometric analysis with normalization to basal expression, and differences among groups were analyzed using two-way ANOVA with Bonferroni post hoc test (*n* = 3, *** *p* ≤ 0.001). The uncropped blots are shown in Appendix A.

**Figure 10 cancers-15-02163-f010:**
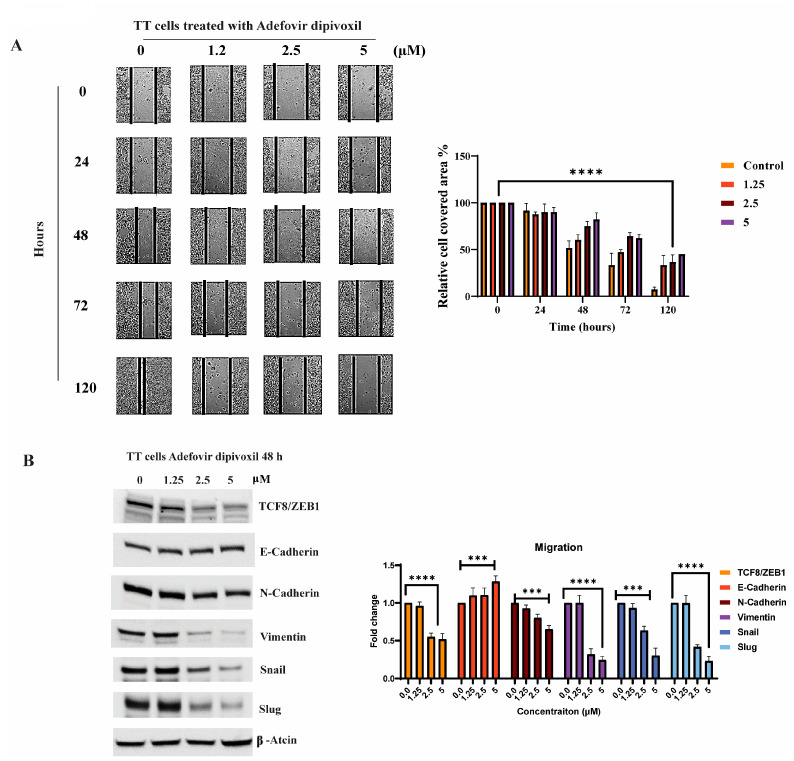
Effects of adefovir dipivoxil on the migration. (**A**) Effect of adefovir dipivoxil in TT cell in wound healing assay. (*n* = 3, *p* ≤ 0.0001) (**B**) Western blotting analyses to selected EMT proteins in TT cells. The western blotting results for TCF8/ZEB1, E-cadherin, N-cadherin, Vimentin, Snail, and Slug were quantified by densitometric analysis with normalization to basal expression, and differences among groups were analyzed using one-way ANOVA (*n* = 3, *** *p* ≤ 0.001, or **** *p* ≤ 0.0001). The uncropped blots are shown in Appendix A.

**Figure 11 cancers-15-02163-f011:**
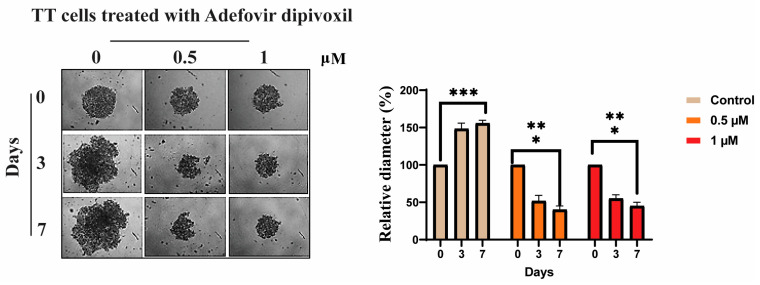
Effects of adefovir dipivoxil on spheroid formation. Spheroids were formed by planting approximately 7000 cells and allowing them to grow for two days before treating them with adefovir dipivoxil. The treatment with adefovir dipivoxil began on day 0 after the initial two-day growth period. Images were taken of spheroids treated with 0, 0.5, and 1 µM of adefovir dipivoxil and their growth and reduction were tracked over a 7-days period. The relative diameter of the spheroids was measured using ImageJ and the results showed a statistically significant effect of adefovir dipivoxil on spheroid growth (* *p* < 0.05, ** *p* < 0.01 and *** *p* < 0.001).

**Figure 12 cancers-15-02163-f012:**
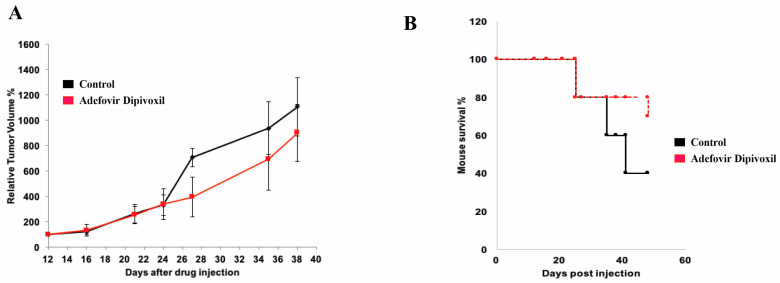
In vivo effect of adefovir dipivoxil. (**A**) Effect of Adefovir dipivoxil on the MTC tumor growth in vivo. (**B**) Effect of Adefovir dipivoxil on the overall survival of MTC tumor bearing mice.

**Table 1 cancers-15-02163-t001:** SP and XP docking scores and MMGBSA binding free energy calculations of adefovir dipivoxil and adefovir against STAT3 SH2 domain.

Name	Standard-Precision (SP) Scoring Function			Extra Precision (XP) Scoring Function		
	Glide Score	Glide Energy (Kcal/mol)	MMGBSA Binding Free Energy (Kcal/mol)	Glide Score	Glide Energy (Kcal/mol)	MMGBSA Binding Free Energy (Kcal/mol)
Adefovir dipivoxil	−3.760	−47.268	−61.39	−4.037	−48.464	−61.23
Adefovir	−6.173	−41.017	−32.11	−5.680	−39.024	−28.30

## Data Availability

All data generated or analyzed during this study are included in this published article.

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
