# Peer review of "Adefovir Dipivoxil as a Therapeutic Candidate for Medullary Thyroid Carcinoma: Targeting RET and STAT3 Proto-Oncogenes"

_cancers, 2023, doi:10.3390/cancers15072163_

Round 1

Reviewer 1 Report (Previous Reviewer 1)

The authors provided their point-by-point responses and clarified some points. However, some of the responses are still necessary to be further clarified before final decision.

Comments:

1) It has been previously reported that RET inhibition causes STAT3 inhibition by its specific phosphorylation at Ser727 in medullary thyroid cancer TT cells (PMID: 34207842). I indeed agree that Adefovir Dipivoxil might specifically target pSTAT3-tyr705 in thyroid cancer, however, it is still notclarified the issue of Ser727 phosphorylation in this study. I recommend authors to either show the results of pSTAT3-ser727 with their treatment or extensively discuss the role of pSTAT3-ser727 in their experimentsIt will fully contribute the regulation of RET and STAT3 in thyroid cancer and the field.

2) Adefovir Dipivoxil treatment significantly inhibited TT cell viability. Inhibited cell viability raises some questions about cell migration. It is well known that inhibition of cell viability may cause cell migration inhibition because cells mainly stop their biological events almost for all stages. Therefore, it is obvious to expect cell migration inhibition. But according to the authors descriptions, Adefovir Dipivoxil specifically targets cell migration. How the authors can make sure whether inhibited cell migration is caused by inhibition of cell viability or not? 

3) Grammar and spelling mistakes were still shown throughout the manuscript. Please revise all with care.

Author Response

For Reviewer 1

we would like to express our sincere gratitude for your invaluable feedback, which has greatly contributed to strengthening our paper. Your insightful comments have provided us with the opportunity to refine and enhance our manuscript.

We are currently addressing each of your questions point by point in order to ensure that our paper thoroughly and accurately responds to your concerns. We believe that incorporating your suggestions will significantly improve the overall quality of our work.

  • It has been previously reported that RET inhibition causes STAT3 inhibition by its specific phosphorylation at Ser727 in medullary thyroid cancer TT cells (PMID: 34207842).I indeed agree that Adefovir Dipivoxil might specifically target pSTAT3-tyr705 in thyroid cancer, however, it is still notclarified the issue of Ser727 phosphorylation in this study. I recommend authors to either show the results of pSTAT3-ser727 with their treatment or extensively discuss the role of pSTAT3-ser727 in their experiments. It will fully contribute the regulation of RET and STAT3 in thyroid cancer and the field.

We pointed out the limitation of our study is we didn’t extensively explore STAT 3 pathway and JAK/STAT.

As suggested by the reviewer, we added this to the manuscript “In addition to the activation of RET and STAT3, it has been noted that CDK5 regulates STAT3 in medullary thyroid cancer, influencing cell proliferation [77]. Suppressing both STAT3 and CDK5 has been shown to slow down human medullary thyroid cancer cell proliferation [77]. Furthermore, overexpression of a mutant STAT3 (Ser 727 to Ala) has a negative impact on the CDK5 pathway and inhibits TT cell growth, highlighting the sig-nificant role of STAT3 Ser 727 and the CDK5 pathway in MTC treatment [37, 77]. The roles of CDK5 and STAT3 were not extensively explored in this study. We believe that further investigation of these roles is necessary for a deeper understanding of the interplay be-tween STAT3, RET, and the CDK5 pathway.”

  • Adefovir Dipivoxil treatment significantly inhibited TT cell viability. Inhibited cell viability raises some questions about cell migration. It is well known that inhibition of cell viability may cause cell migration inhibition because cells mainly stop their biological events almost for all stages. Therefore, it is obvious to expect cell migration inhibition. But according to the authors descriptions, Adefovir Dipivoxil specifically targets cell migration. How the authors can make sure whether inhibited cell migration is caused by inhibition of cell viability or not? 

In epithelial-mesenchymal transition (EMT), cells typically exhibit enhanced migration, increased drug resistance, and a reduced rate of proliferation until they undergo mesenchymal-epithelial transition (MET) [PMID: 30728292 / PMID: 24202173/ PMID: 31514467]. In our study, we employed a scratch assay to demonstrate the inhibition of cell migration. Furthermore, as a confirmatory experiment, we assessed the expression of EMT-related proteins (such as ZEB1, E-cadh, N-cadh, Vimentin, snail and slug), which serve as reliable indicators of the cellular reprogramming into the EMT process. The findings from both analyses suggest a reduction in migration and invasion, supporting the inhibitory effect of adefovir dipivoxil. “PMID: 19726629 / PMID: 19546857 / PMID: 19951899 PMID: 33777948”

  • Grammar and spelling mistakes were still shown throughout the manuscript. Please revise all with care.

Thank you for pointing out the grammar and spelling mistakes in our manuscript. We have carefully reviewed and revised the entire manuscript to address these issues. To help you track the changes, we have utilized the 'Track Changes' feature in Microsoft Word, which will allow you to easily identify and review the corrections we have made.

Reviewer 2 Report (Previous Reviewer 3)

It is hard to piece together original Western blot images. Align the different blots of the same gel, label the lines and the protein detected and clearly indicate the different experiments referred to each figure (experiment 1, 2, 3).

Clearly indicate differences between adefovir dipivoxil and adefovir STAT3 binding pockets in Figure 5B.

The higher effect of adefovir dipivoxil obtained on TT cell proliferation is related to the inhibition of RET expression and RET signaling pathway. On the other hand, the effect of adefovir dipivoxil on the DNA damage is not specific for RET mutated cell lines, being dissociated by RET inhibition, and could account for the effect observed in other cell lines.

Considering that the authors did not characterize stem cell markers in spheroids, the authors should change CSC with 3D growth in the text.

As suggested by the authors, the less inhibitory effect obtained in MZ-crc-1 with adefovir dipivoxil may be due to the additional mutations present in this cell line, in particular in PI3K. The authors should test the effect of adfovivir dipivoxil on RET expression and PI3K signaling pathway in MZ-crc-1 cells.

Author Response

Please see attachment , 

Round 2

Reviewer 2 Report (Previous Reviewer 3)

The authors' replies are quite satisfying but should be incorporated in the text:

Statement at lines 354-8 should be evident in figure 5. 

The lack of effect on RET protein expression in MZ-CRC-1 cells indicates that adefovir dipivoxilthe Is inactive on RET M918T. This finding should be shown and discussed.

It Is unknown whether the effect of adefovir dipivoxilthe on DNA damage is predominant in TT cells, since ATR activation were not investigated in other cell lines. On the other end, differences in cell proliferation and viability may be consequnce of inhinition of RET signaling rather than DNA damage. Discuss this point.

Author Response

Comments and Suggestions for Authors

The authors' replies are quite satisfying but should be incorporated in the text:

Thank you for taking the time to review our manuscript. We appreciate your valuable feedback and suggestions. We have taken your comments into consideration and have incorporated the author's replies into the text as you suggested. We believe this will provide a clearer and more comprehensive understanding of the information for the reader. We also added a supplementary figure which can help the reader to go through the manuscript.

Statement at lines 354-8 should be evident in figure 5. 

The statement was added to the legends as requested. We also add a supplementary figure 2S for further clarification.

The lack of effect on RET protein expression in MZ-CRC-1 cells indicates that adefovir dipivoxilthe Is inactive on RET M918T. This finding should be shown and discussed.

We have added the results for MZ-CRC-1 M918T as a supplementary figure S1. (attached)

Also, in the results section, we added "Moreover, the level of the RET protein in the MZ-CRC-1 cell line, which harbors a mutated RET gene (M918T), did not alter after being treated to different concentrations of adefovir dipivoxil for 48 hours, as shown in Figure S1 (supplementary figures)."

In the discussions section, we added this sentence with the references: "Although MZ-CRC1, an MTC cell line with the M918T mutation, appears to be more resistant to adefovir dipivoxil treatment compared to TT cells carrying the C634W mutation (Figure S1), it can be hypothesized that specific genes might be highly overexpressed in the RET M918T mutant but not in the C634W mutant. This overexpression could contribute to the increased resistance of MZ-CRC1 to the treatment [29-60].

It Is unknown whether the effect of adefovir dipivoxil the on-DNA damage is predominant in TT cells, since ATR activation were not investigated in other cell lines. On the other end, differences in cell proliferation and viability may be consequence of inhibition of RET signaling rather than DNA damage. Discuss this point.

We added a paragraph for this comment: “The effect of adefovir dipivoxil on TT cells may be related to the combination of inhibiting the RET signaling pathway and ATR activation. This could explain the diverse effects of adefovir on other cell types that are not driven by RET-related cancer, RET/PTC1, and RET M918T. Additional studies are needed to gain a deeper understanding of the mechanisms involved in ATR-mediated DNA damage responses and how they differ among distinct cancer cells [81-82].

This manuscript is a resubmission of an earlier submission. The following is a list of the peer review reports and author responses from that submission.

Round 1

Reviewer 1 Report

Based on my previous comments, although the authors did not provide point by point answers, I see many improvements in this manuscript. I am appreciated that they repeated their western blot data and mRNA expression data as well as statistical analysis. However, I still have same questions which is not answered by authors yet, and it needs to be clarified before publishing in “Cancers”.

Major revisions/questions:

1) They mentioned STAT3 inhibition in TT cells by only showing pSTAT3-tyr705. However, I would like to see the effect of Adefovir-dipivoxil on pSTAT3-Ser727 as well. Because, it has been well established that pSTAT3-Ser727 also plays essential role in medullary thyroid cancer progression. Please see the articles about pSTAT3-Ser727 in TT cells (PMID: 17145757) and cite these two papers in the main text (line 65, and 334 are appropriate to cite these papers). In addition, I recommend to authors that it would be better to write additional section in discussion part about why the authors did not focus pSTAT3-Ser727.

2) The authors also showed that mTOR and AKT inhibition (RET-regulated PI3K/AKT/mTOR signaling is well known). In my opinion, the authors should mention about how and why they choose this protein as downstream of RET protein under Adefovir-dipivoxil treatment since there is multiple downstream signaling axis controlled by RET mutations. The signaling mechanism provided by authors is not clear. Because they provided multiple signaling axis such as EGR1, CREB; STAT3, RET; and PI3K/AKT/mTOR. what is the connection between all signaling axis in TT cells? Importance and physiological functions of these signaling axis in TT cells. I recommend that the authors should mention this point more detailly and discuss it in the discussion section.

Minor revision:

**Line 51, and others references in the main text: Merge the citations. For exp. (not true: [3], [4]. True: [3,4]).

**Title of results and figures should be rewritten as present tense, and cover all findings in a sentence.

**Figure 5: If it is possible to enhance resolution, it would be better to see the results

Author Response

Based on my previous comments, although the authors did not provide point by point answers, I see many improvements in this manuscript. I am appreciated that they repeated their western blot data and mRNA expression data as well as statistical analysis. However, I still have same questions which is not answered by authors yet, and it needs to be clarified before publishing in “Cancers”.

We would like to apologize for not thoroughly addressing the questions in our previous submission. We appreciate the opportunity to answer them now and make the necessary changes in the main text.

Major revisions/questions:

  • They mentioned STAT3 inhibition in TT cells by only showing pSTAT3-tyr705. However, I would like to see the effect of Adefovir-dipivoxil on pSTAT3-Ser727 as well. Because, it has been well established that pSTAT3-Ser727 also plays essential role in medullary thyroid cancer progression. Please see the articles about pSTAT3-Ser727 in TT cells (PMID: 17145757) and cite these two papers in the main text (line 65, and 334 are appropriate to cite these papers). In addition, I recommend to authors that it would be better to write additional section in discussion part about why the authors did not focus pSTAT3-Ser727.

  • Canonical STAT3 activation requires phosphorylation of Y705, resulting in STAT3 dimerization, nuclear translocation, and binding to a DNA. Due to the dimerization process of STAT3 (and other STAT) proteins, many small peptides or peptidomimetics, as well as small molecule inhibitors, have been created. These target the interaction between the SH2 domain and the phosphorylated Y705 residue which is our focus on the adefovir dipivoxil (PMID: 30578415).

  • From the recommended article titled 'Cdk5 regulates STAT3 activation and cell proliferation in medullary thyroid carcinoma cells', it was stated that medullary thyroid cancer patients with RET germline mutations exhibit higher STAT3 activation and nuclear localization. In this particular sentence, they are referring to STAT3 Y705. The reference used for this statement is PMID: 30578415.

  • We would like to inform the reviewer that we have addressed their question by implementing the recommended reference in the text, specifically at line #337-340 (reference #37). Additionally, we have added a new paragraph (line #629-633) to explain the use of STAT3 Y705 and its significance in the context of our study. This paragraph is also supported by reference #73.

  • The authors also showed that mTOR and AKT inhibition (RET-regulated PI3K/AKT/mTOR signaling is well known). In my opinion, the authors should mention about how and why they choose this protein as downstream of RET protein under Adefovir-dipivoxil treatment since there is multiple downstream signaling axis controlled by RET mutations. The signaling mechanism provided by authors is not clear. Because they provided multiple signaling axis such as EGR1, CREB; STAT3, RET; and PI3K/AKT/mTOR. what is the connection between all signaling axis in TT cells? Importance and physiological functions of these signaling axis in TT cells. I recommend that the authors should mention this point more detailly and discuss it in the discussion section.

  • The PI3K/AKT/mTOR and ERK pathways are active in preclinical models and cases of medullary thyroid carcinoma. There is much evidence suggesting that PI3k/mTOR/AKT and RAS/MEK/ERK have a pivotal role in initiating tumors (PMCID: PMC4707207). Drugs such as everolimus, an mTOR inhibitor, have shown effectiveness in treating MTC patients, but response rates have not been high due to oncogene cooperativity. This suggests the importance of the PI3K/AKT/mTOR pathway in MTC as a therapeutic target. In preclinical studies, inhibiting both the RAS/MEK/ERK and PI3K/AKT/mTOR pathways has shown promise in treating MTC, but the use of combination drugs is limited due to toxicity (PMID: 21543427, PMID: 25115638). Adefovir dipivoxil has shown inhibitory activity in both pathways in vitro, and its toxicity is reversible upon discontinuation of the drug, as we have pointed out in the manuscript.

  • To address the reviewer comment, we have included the following in the text (In this study, we investigated how adefovir dipivoxil reduces the proliferation of TT cells by downregulating RET expression. The oncogenic RET activation promotes cell growth and survival by activating various intracellular signaling pathways. In a previous study by Drosten et al., they characterized the downstream signaling pathways associated with RET required for tumor progression and maintenance in TT cells [55*]. They disrupted RET phosphorylation and activation in TT cells by using an adenoviral vector expressing the dominant-negative truncated RET protein that lacks the intracellular tyrosine kinase domain. This resulted in downregulation of the Raf/MEK/ERK and PI3K/Akt/mTOR pathways, suggesting that these pathways are mainly involved in RET-mediated transformation. Consistent with this study, our results showed that adefovir dipivoxil's suppression of RET expression inhibited the phosphorylation of ERK, AKT and mTOR, and further decreased the expression of cyclin D1, P53 and Bcl-2, which are regulated by the Raf/MEK/ERK and PI3K/Akt/mTOR pathways (Figure 8A). Notably, we also found that normal thyroid cells, Nthy-ori-3-1, and hepatocytes cell lines showed significant resistance to adefovir dipivoxil indicating that this compound is more selective for MTC)

Line #570-585

Minor revision:

**Line 51, and others references in the main text: Merge the citations. For exp. (not true: [3], [4]. True: [3,4]).

Edited.

**Title of results and figures should be rewritten as present tense, and cover all findings in a sentence.

Edited

**Figure 5: If it is possible to enhance resolution, it would be better to see the results.

Please see the attachment PDF for better resolution.

Reviewer 2 Report

The authors added statistical analysis of their experiments which I am happy with. 

Reviewer 3 Report

In this revised version of the paper the authors did not provide a point-by-point responses and many comments were not addressed. In particular:

In the row data, the authors provided partial gel images relating to only one experiment, not three.

In Figure 2D, adefovir dipivoxil seems to reduce the level of ERK rather than pERK.

Adefovir dipivoxil is metabolized in the active metabolite adenfovir. Therefore, I do not see the rationale of investigating differences in Stat3 interaction between the two molecules at paragraph 3.5. Besides, differences between adefovir dipivoxil and adefovir STAT3 binding pockets were not evident in Figure 5B.

The authors did not directly demonstrate that adefovir dipivoxil causes apoptosis, as stated in the abstract. 

As hypothesized by the authors (lines 359-362), the effect of adefovir dipivoxil on DNA damage and cell cycle is due to its binding to DNA and is independent from Ret signaling inhibition (in contrast with statement at lines 350-352). These mechanisms are therefore not specific for TT cells (Voss et al. 2021 Front immunol) and may be associated to side effects.

The formation of spheroids is not an indicator of CSC without stem cell markers characterization but rather represent a 3D growth. Moreover, results in spheroids and mice were not discussed.

In addition:

RET M918T is the most frequent mutation in sporadic MTC. Therefore, the less pronounced inhibitory effect of adefovir dipivoxil in Mz-CRC-1 cell line may represent a problem.

Limitations should be claimed in the Discussion, not in the Results.

In Figure 8A the basal expression of cyclin D is not represented in the graph.

Author Response

In this revised version of the paper the authors did not provide a point-by-point responses and many comments were not addressed. In particular:

We apologize for not fully addressing the questions in our previous submission and thank you for giving us another chance to provide the appropriate responses. We will now answer the questions and make the necessary revisions to the main text accordingly.

In the row data, the authors provided partial gel images relating to only one experiment, not three.

We added the 3 or more experiment for each western blot. File attached.

In Figure 2D, adefovir dipivoxil seems to reduce the level of ERK rather than pERK.

This is correct. We have observed a small decrease in total ERK expression, especially with the p42 band. However, whenever we use pERK/ERK, we always find that the level of pERK is lower than that of ERK, as highlighted in the statistical analysis.

Adefovir dipivoxil is metabolized in the active metabolite adenfovir. Therefore, I do not see the rationale of investigating differences in Stat3 interaction between the two molecules at paragraph 3.5. Besides, differences between adefovir dipivoxil and adefovir STAT3 binding pockets were not evident in Figure 5B.

Adefovir dipivoxil is converted to adefovir when given orally and then to Adefovir MP –> Adefovir DP. As seen above (PMID: 15450948), all four compounds have slightly different structures. The main purpose of this study is to administer adefovir dipivoxil parenterally to bypass first-pass metabolism and evaluate its activity. These structural differences may have varying impacts on drug binding and efficacy.

The authors did not directly demonstrate that adefovir dipivoxil causes apoptosis, as stated in the abstract. 

We demonstrated the activation of the intrinsic apoptosis pathway in TT cells by evaluating the levels of BCL2 and P53 protein expression. Additionally, we measured caspase 3 activity and observed an increase in activity, as shown in the figure below (these data were not included in the manuscript).

As hypothesized by the authors (lines 359-362), the effect of adefovir dipivoxil on DNA damage and cell cycle is due to its binding to DNA and is independent from Ret signaling inhibition (in contrast with statement at lines 350-352). These mechanisms are therefore not specific for TT cells (Voss et al. 2021 Front immunol) and may be associated to side effects.

The effect on the RET signaling pathway and cell cycle protein is more specific to TT cells, as observed in the western blot and cell proliferation (IC50) assays. Other cell lines showed higher IC50 values compared to TT cells, as depicted in Figure 3. Furthermore, the effect of adefovir dipivoxil requires the presence of RET mutation, as shown in the figures below (these figures are not included in the manuscript). However, we found that the DNA damage checkpoint was not related to the RET signaling pathway, as demonstrated by the use of a CHK1 inhibitor (Fig.9B). We have revised the sentences in the manuscript accordingly.

The formation of spheroids is not an indicator of CSC without stem cell markers characterization but rather represent a 3D growth. Moreover, results in spheroids and mice were not discussed.

  • It has been established that EMT (epithelial-to-mesenchymal transition) has a crucial part in the spreading and reappearance of tumors, and this connection is strongly associated with the activity of CSC (cancer stem cells). (PMID: 18485877/ PMID: 26284927 / PMID: 26328525/ PMID: 26122848 / PMID: 21640118)
  • Spheroid formation (EMT) and in vivo data have been included in the discussion (Line 659-690)

In addition:

RET M918T is the most frequent mutation in sporadic MTC. Therefore, the less pronounced inhibitory effect of adefovir dipivoxil in Mz-CRC-1 cell line may represent a problem.

  • Mz-crc-1 carrying the M918T mutation is a different model than TT cells carrying the C634R mutation. Multiple drugs have shown efficacy in either or both cell lines. For further clarification, SU5402 has demonstrated higher efficacy on TT cells than MZ-CRC (PMID: 36139603). This finding supports our hypothesis that TT cells are more sensitive to adefovir dipivoxil.
  • Additionally, it has been reported that MZ-Crc has a missense mutation in PIK3CA and a mutation in MAX (for more information, see PMID: 30737244)

Limitations should be claimed in the Discussion, not in the Results.

We have added the necessary limitations of our study, such as mentioning the limitation of RT-PCR immediately after presenting the results to provide a better understanding to the reader. We have also included the limitation of adefovir concentration in vivo in the discussion section. This has been done to maintain the flow of the discussion and avoid creating fragmented sentences that are not related to the main idea.

In Figure 8A the basal expression of cyclin D is not represented in the graph.

fixed
